# Primary Bone Lesions in Rosai–Dorfman Disease, a Rare Case and Diagnostic Challenge—Case Report and Literature Review

**DOI:** 10.3390/diagnostics12040783

**Published:** 2022-03-23

**Authors:** Razvan Adam, Tudor Harsovescu, Sorin Tudorache, Cosmin Moldovan, Mark Pogarasteanu, Adrian Dumitru, Carmen Orban

**Affiliations:** 1Department of Orthopedics and Traumatology, Elias Emergency University Hospital, 011461 Bucharest, Romania; 2Department of First Aid and Disaster Medicine, Titu Maiorescu University, 040051 Bucharest, Romania; 3Department of Preclinical Sciences, Anatomy and Embryology, Titu Maiorescu University, 040051 Bucharest, Romania; secretar_stiintific@yahoo.com; 4Department of Clinical Sciences, Surgery, Titu Maiorescu University, 040051 Bucharest, Romania; moldovan.cosmin@gmail.com; 5Department General Surgery, Clinical Hospital No.1 CF Witting, 010243 Bucharest, Romania; 6Department of Orthopedics and Traumatology, Carol Davila University of Medicine and Pharmacy, 020021 Bucharest, Romania; mark.pogarasteanu@gmail.com; 7Department of Orthopedics and Traumatology, Dr. Carol Davila Central Military Emergency University Hospital, 010825 Bucharest, Romania; 8Department of Pathological Anatomy, Carol Davila University of Medicine and Pharmacy, 020021 Bucharest, Romania; dr.adriandumitru@yahoo.com; 9Pathoteam Diagnostics, 051923 Bucharest, Romania; 10Department of Anesthesia and Intensive Care, Carol Davila University of Medicine and Pharmacy, 020021 Bucharest, Romania; carmen.orban@spitalulmonza.ro; 11Intensive Care Unit Department, Monza Oncology Hospital, 013812 Bucharest, Romania

**Keywords:** Rosai–Dorfman disease, bone involvement, rare case, atypical evolution, difficult diagnostic

## Abstract

Rosai–Dorfman Disease (RDD), also known as sinus histiocytosis, is included in the group of rare diseases, characterized by proliferation and accumulation of histiocytes in the lymph nodes (lymphadenopathy), most often involving the cervical ganglion chains (nodal form). RDD bone involvement is rare, estimated at 10% of cases, but primary bone involvement (extranodal form), is very rare—2–8%. Usually they are solitary lesions, with multifocal primary bone manifestations being extremely rare. Histopathological analysis is of high value for a correct diagnosis. We present the case of a Caucasian woman, 42 years old, initially treated in another clinic, for an osteolytic tumor formation in the right tibial shaft. An excisional biopsy with bone trepanation was performed, the histopathological diagnosis being the chronic inflammatory tissue. The evolution was atypical, with tumor growth, extraosseous, subcutaneous. A needle biopsy was repeated in our clinic, the result being similar to the original one. Evolution of the tumor, and the radiological and imaging aspect (periosteal reaction, eroded and thin bone cortex) suggested a more aggressive disease, these being in inconsistency with the result obtained. The biopsy was repeated, as an excision type this time. The histopathological result and immunohistochemistry indicated an RDD primary bone lesion. Based on this result, and corroborated with the data from the literature, we initiated the surgical treatment, curettage and grafting with bone substitute plus safety osteosynthesis with locked plaque, the patient registering a favorable evolution. RDD primary bone lesions are in fact an atypical manifestation of a rare disease. The correct diagnosis is very difficult due to the non-specific imaging aspect. Histopathological examination errors, especially in the case of needle biopsies can lead to errors in diagnosis and treatment with negative results for the patient.

## 1. Definition and Classification

Rosai–Dorfman, also known as sinus histiocytosis, is included in the group of rare diseases, characterized by proliferation and accumulation of histiocytes, cells from macrophage-dendritic lineage, in the lymph nodes (lymphadenopathy), most often involving the cervical ganglion chains, cervical lymphadenopathy.

However, the accumulation of histiocytes in other organs, called extranodal involvement, is not uncommon. It may occur in more than 40% of patients, sometimes without associated lymphadenopathy [1,2]. Cutaneous tissue (10%), nasal cavity and paranasal sinuses (11%), eye and orbits (11%) and the central nervous system (5%, predominantly dural) are the most frequently affected extra nodal sites [3,4], and in some cases, synchronous involvement of multiple extranodal was noticed. On the other hand, bone lesions are rare in RDD, occurring in 5 to 10% of cases [5], and even rarer are RDD primary bone lesions without lymphadenopathy—2–8%. Several cases have been reported in the literature, the largest series being 15 cases [6].

This disease was described in 1969 by Rosai and Dorfman, being classified by the Working Group of the Histiocyte Society of 1987 as a non-Langerhans cell (LC) histiocytosis [3]. This was defined by the accumulation of histiocytes that do not meet the phenotypic criteria for the diagnosis of Langerhans cells (LCs) [7]. Most recently a revised classification of histiocytic disorders and neoplasms of the macrophage-dendritic cell lineage was proposed, in which Rosai–Dorfman disease (RDD) gets to form its own subtype (“R group”) due to its unique characteristics [8]. Usually, cell proliferation of various histiocytic diseases can be either malignant or non-malignant; however, in RDD, histiocyte proliferation is likely reactive and polyclonal [9]. This “R group” of histiocytoses, includes familial RDD, sporadic RDD and other miscellaneous non-cutaneous, non-LC histiocytosis [10]. Cutaneous RDD demonstrates unique epidemiological and clinical features and therefore was classified separately from other forms of RDD [3], as part of the ‘C group’ of histiocytosis [10], and it was further divided into xanthogranuloma and non-xanthogranuloma families, with cutaneous RDD being part of the last one [11]. Distinction from the xanthogranuloma family (S100-negative) is an important differential diagnosis.

Diseases that were classified as part of the “R group” of histiocytosis are summarized in Figure 1 [3]. The most common form is Sporadic RDD including the classic nodal form, extra nodal RDD, neoplasia-associated RDD and immune disease-associated RDD [10].

**Clinical features:** In most cases, painless swelling or enlargement of affected lymph nodes, most common cervical one (cervical lymphadenopathy), can occur, affecting adolescents and young adults [12]. In some cases, fever, loss of weight and night sweats may be associated [13]. It occurs more commonly in African patients with a slight male predominance, male to female ratio of 1.4 [5]. Central nervous system involvement may mimic meningioma clinically and is usually not associated with nodal disease [14].

As it was mentioned earlier, bone involvement is rare, estimated at 10% of cases [15] the clinical signs being local pain and swelling, associated with joint pain if the lesion is located in the metaphyseal or epiphyseal area. At the same time, bone lesions may also be found incidentally. In the case of primary bone disease, most often, solitary lesions are found, multifocal primary bone manifestations being extremely rare. Primary osseous RDD typically is solitary and has been reported in the femur, tibia, skull, clavicle, sacrum, and small bones of the hands and feet [16].

**Paraclinical features:** Increased laboratory inflammation markers such as ESR, Fibrinogen, C-reactive protein, may exist but are not mandatory.

Histopathological analysis is of high value for a correct diagnosis. Sinus dilatation due to typical histiocyte proliferation is noticed in lymph node histology [4], being enlarged and matted together, forming firm multinodular masses with a yellow–white appearance [17]. At the level of the sinusoids, numerous large histiocytic cells can be observed, with contoured smooth hypochromatic nuclei, small, different round nucleoli, placed centrally and poorly defined, pale, fragile cytoplasm [5]. Emperipolesis, a useful feature (engulfment of intact lymphocytes contained with the cytoplasm of histiocyte cells), is seen as intact hematolymphoid cells within a vacuole or floating freely in the cytoplasm of the histiocytes [3].

The histology of extraganglionic disease is similar to ganglionic RDD but may have more prominent lymphoid follicles with germinal centers, fibrosis, sclerosis, fewer histiocytes and more subtle emperipolesis [3]. The histiocytes are S100, CD68 and CD163 positive and are by definition CD1a and langerin (CD207) negative, excluding LCH in the differential diagnosis. The S100 stain often highlights the emperipolesis [3]. The Histiocyte Society recommends that all cases of RDD should be evaluated for IgG4-positive plasma cells (grade D2 evidence) because the plasma cell population in the cortex can demonstrate abundant IgG4-positive plasma cells and at the same time will stain with plasma cell markers (CD38, CD138 and MUM1) [3].

The radiological and imaging manifestations of bone RDD are not specific to the disease. In the case of bone lesions, CT imaging, MRI and radiological investigations show lytic lesions, most often, with well defined, sclerotic margins. Pure sclerotic lesions have been reported but are very rare being even considered unusual [18]. Internal osseous septations or calcified matrix can be present. Periosteal reaction can be found, but it is also rare. Cortical thinning and focal breakthrough are common findings. Soft tissue mass in continuity with the bone lesion is a rare finding. Based on these observations, the differential diagnosis can be made with chronic inflammatory or infectious bone lesions, lytic bone metastases or bone sarcoma, especially in the presence of the periosteal reaction.

**RDD primary bone involvement treatment:** In the case of treatment there are two options. Since sometimes the symptoms are minimal or non-existent and the bone damage is not extensive, only a close observation of the case is recommended, spontaneous remission being possible, as Pulsoni shows in his study [19].

Regarding the surgical treatment, depending on the damage to the bone structure, one can opt for block resection and reconstruction or curettage and grafting of the remaining bone cavity.

## 2. Case Report

We present the case of a 42-year-old Caucasian woman who came to our clinic with pain and regional deformity located in the distal third of her right calf.

About 6 months ago, the patient presented to another orthopedic clinic complaining of pain in the distal half of the right calf, with sporadic intensification, accentuated by physical overload, sometimes of a pulsating nature, associated with episodes of swelling of the calf.

Next, the patient was investigated by radiology, computer tomography (CT) imaging and bone scintigraphy. Radiological examinations (Figure 2) and contrast-enhanced computed tomography (Iomeron) describe an osteolytic lesion located in the distal third of the right tibial shaft with dimensions of 37/22/16 mm, associating thinning of the cortex with minimal interruptions and periosteal reaction on the anterior and internal face of the tibia. Contrast-enhanced CT examination at the abdominal, thoracic and pelvic levels did not reveal any other lesions of the examined organs, except for a benign cyst located in the superior external quadrant of the left breast.

Bone scintigraphy (99 m Tc) Tc HDP at a dose of 740 MBq/5.92 mSv, administered intravenously, showed in the vascular phase, an increased vascularization in the distal third of the calf and in the immediate and late phases—heterogeneous hyper captive area in the distal third of the right tibia.

Whole-body bone scintigraphy does not show other hyper captive lesions, except for a minimal capture at the level of the bilateral shoulder joints, on an inflammatory background. The conclusion of this investigation was: active metabolic process located in the distal third of the right tibia, visible in all three phases

Unfortunately, CT scan and scintigraphy images could not be obtained from the other clinic, only their interpretations.

Based on these investigations, they decided to perform an excisional biopsy with trepanation of the antero-internal tibial bone cortex at the level of the tumor formation. The result of the histopathological examination (hematoxylin–eosin staining) and immunohistochemistry (AE1-AE3, ACT, CD68, CD20, CD3, Ki67) was: a chronic intraosseous inflammatory process with areas of regenerative appearance at the periphery.

At the time of presentation in our clinic, during the clinical examination, subcutaneous, extraosseous development of a tumor formation was observed. The tumor was located on the antero-internal face of the leg, corresponding to the level of the previous biopsy. The formation had a firm, elastic consistency, on palpation, adherence to the bony plane, non-adherence to the cutaneous plane, with well-defined edges. Palpation induced local pain. The general clinical examination did not reveal any other symptoms.

Corroborating the clinical examination with the patient’s history, we had decided to investigate again radiological and imaging (CT, MRI). Radiological and CT examination showed a slight increase in the size of the tumor formation, confirmed by MRI images (Figure 3), at this time the dimensions being 5.5/1.95/2.2 cm. Radiological and CT features were similar, osteolytic lesion with thinning and disruption of the bone cortex, associating periosteal reaction (Figure 4). However, this time, the CT images show the extraosseous expansion of the tumor formation, through the bone window created by the initial biopsy (Figure 5).

MRI examination clearly shows the extraosseous, subcutaneous expansion of the tumor formation through the cortical bone defect. This extraosseous expansion is in the continuity of the intraosseous tumor tissue, with the same imaging characteristics, having dimensions of 2.92/3.2/1.5 cm and well-defined edges, encapsulated (Figure 6).

The obtained results showed a surprising, atypical evolution of the tumor, compared to the initial histopathological result, chronic inflammatory lesion.

Laboratory inflammation markers showed elevated values, ESR: 29 mm/h (NV 20 mm/h), fibrinogen 485 mg/dL (NV180–450 mg/dL) serum C-reactive protein CRP 6.44 mg/L (NV < 5 mg/L). Alkaline phosphatase laboratory values were within normal limits 77.92 U/L (NV 35–104 U/L).

These could support the histopathological diagnosis of chronic inflammatory lesions, but the clinical evolution with extracompartmental development of the tumor and the imaging, thinning and disruption of the bone cortex in association with periosteal reaction, suggested the presence of a more aggressive lesion than the one initially diagnosed, which is why we decided to repeat the biopsy. A percutaneous biopsy, on a Best Lisas biopsy needle, was performed, collecting samples for both histological and bacteriological examination. This time too, the result of the histopathological examination was a chronic inflammatory lesion, but xanthogranulomatous cells were also identified. The bacteriological results were negative; no germs were identified in the harvested tissue.

This new histopathological result has only accentuated the diagnostic confusion, not corresponding to the radiological and imaging aspects and to the atypical clinical evolution. Based on the clinical and paraclinical data that we had, the differential diagnosis could be made with the following conditions.

Adamantinoma: it is a rare malignant bone neoplasm of epithelial origin with preferential location in the tibial shaft (80%). Radiological, osteolytic lesions with peripheral condensation and thinning of the bone cortex, but without periosteal reaction. Soft tissue involvement is usually associated with advanced or recurrent lesions. Inflammatory markers are within normal limits. Histologically it is a biphasic tumor characterized by epithelial and osteofibrous components, sometimes nuclear atypia has been observed. Immunohistochemically, all adamantinomas are uniformly positive for keratins 14 and 19. The epithelial cells show co-expression of keratin, especially basal epithelial cell keratins (CKs 5, 14 and 19) and vimentin.

Xanthoma: pseudotumor lesion, osteolytic, with erosion of the bone cortex, but without periosteal reaction. Histologically, xanthomatous cells are identified in conjunctive tissue. Inflammatory markers are within normal limits.

Osteolytic osteosarcoma: aggressive osteolytic lesion, with thinning and rupture of the bone cortex and extraosseous development. Associate periosteal reaction. Extraosseous development is not well defined and without peripheric capsule. Alkaline phosphatase levels are increased. Histopathological examination identifies specific cells.

Osteolytic bone metastases: osteolytic lesions with thinning and rupture of the bone cortex, but without periosteal reaction. The extraosseous extension is not well defined and does not have a capsule at the periphery. Alkaline phosphatase levels are increased.

All these features are summarized in Table 1.

Thus, we decided to perform a new biopsy, this time excisional, collecting tissue samples from both the bone and the extraosseous tumor tissue. We requested additional histological analyzes of the tumor samples.

The specimen samples were fixated with 10% buffered formalin and were processed by conventional histopathological methods using paraffin embedding, sectioning and hematoxylin–eosin (HE) staining. Microscopic examination showed a diffuse infiltrate of small lymphocytes, plasma cells and frequent foamy histiocytes as well as pigment laden macrophages (siderophages) displayed around small vascular structures with slightly thickened vascular walls (Figure 7). Some histiocytes displayed abundant, pale cytoplasm with phagocytized lymphocytes, suggestive for emperipolesis. This inflammatory infiltrate extended into the adjacent bone and soft tissue (Figure 8). The inflammatory reaction includes rare fragments of devitalized bone lamellae with dystrophic calcifications, found on the periphery of the analyzed specimens.

In order to help establish an accurate final diagnosis, IHC tests were performed. To begin with, the paraffin blocks were cut and the resulting 3-μm thick sections were mounted on slides covered with poly-L-lysine. Afterwards, the sections were deparaffinized in successive toluene and alcohol baths, rehydrated (three successive alcohol baths with decreasing concentration: 96%, 80% and 70%) followed by a final bath with distilled water for 10 min. For IHC staining, we used an indirect tristadial Avidin–Biotin complex method (deparaffinization in toluene and alcohol series), rehydration, washing in phosphate-buffered saline (PBS), incubation with normal serum, for 20 min, incubation with primary antibody overnight, DAKO Labeled Streptavidin–Biotin (LSAB) kit, washing in carbonate buffer and development in 3,3′-diaminobenzidine (DAB) hydrochloride/hydrogen peroxide nuclear counterstaining with Mayer’s Hematoxylin. Immunohistochemical analysis demonstrated that the foamy histiocytes stain intensely for S100, CD68, CD163 and Cyclin D1, while CD1a reaction was negative (Figure 9).

In conclusion, the microscopic aspects corroborated with the immunohistochemical expression pattern are highly suggestive for Rosai–Dorfman Disease.

Due to the rarity of this condition and specifically the bone involvement in the disease, associated with the fact that the patient did not have specific symptoms, we made a review of the scientific literature. Corroborating the data from the literature, presented below, with our clinical data, imaging, laboratory markers and histopathological result, we concluded that indeed, we are dealing with a rare case, extra nodal Rosai–Dorfman disease and primary bone lesions.

Cases reported in the scientific literature with extranodal involvement show only local symptoms, swelling and pain, without other systemic manifestations, such as our case.

Radiological and imaging features from the literature are osteolytic lesions with or without sclerotic rim, with heterogenous signal (MRI), and sometimes internal osseous septations can be identified. Thinning and sometimes disruption of the bone cortex are constantly described. The periosteal reaction may be present, but it is an inconsistent feature. No soft tissue tumors or perilesional edema have been reported.

In our case we identified radiological and imaging features like those in the literature. Osteolytic bone lesion, with thinning and disruption of the bone cortex, with heterogeneous appearance of the tumor on MRI images, without intraosseous septation but associating periosteal reaction.

The extraosseous development of the tumor, through the cortical defect created by the excisional biopsy, represents an atypical feature, for us being in fact, a diagnostic pitfall.

In the cases from the literature, the definite diagnosis was obtained based on the histopathological result, characteristic being emperipolesis (engulfment of intact lymphocytes contained with the cytoplasm of histiocyte cells) and positive S100 immunohistochemical staining. All these elements were also identified in the histopathological analysis of our case.

After establishing the diagnosis, the surgical treatment was initiated. We performed block resection of the extraosseous tumor formation followed by the curettage of the intradiaphyseal lesion, with the removal of the tumor tissue to the level of healthy bone tissue. The curettage process was verified by intraoperative radiology images (C-Arm mobile radiology). Tumor tissue samples were taken for histopathological examination, both from the extraosseous tumor and from the intradiaphyseal tissue, with the result being similar—RD bone lesion. After curettage, we washed the remaining bone cavity with 70% ethyl alcohol to destroy any remaining tumor cells. In order to fill the defect, bone grafting was practiced using tricalcium phosphate bone substitute (β-TCP) 25%/75% hydroxyapatite, irregular granules 3–4 mm in diameter, using about 60 cc.

As bone strength was diminished by tumor erosion of the cortex, there was a risk of fracture. Thus, safety osteosynthesis, using one plate (titanium alloy) with locked screws was associated (Figure 10).

The immediate postoperative evolution was marked by the appearance of an area of superficial skin dehiscence, in the area where the extraosseous tumor developed, the area where the skin was thin, poorly vascularized. We performed local debridement and sutured per secundam, using in addition, adhesive tapes to reduce skin tension so as not to disturb the local vascularization. The evolution was favorable, with the skin healing (Figure 11).

The subsequent evolution was favorable, 6 months after the surgery the patient no longer having local pain. The radiological examination showed a favorable local evolution, without signs of tumor recurrence (Figure 12). Furthermore, the patient did not show any other specific symptoms of RDD, which is why no systemic treatment (glucocorticoids or other immunomodulatory drugs) was initiated.

## 3. Literature Review

Given the rarity of primary bone involvement cases in RDD and the difficulty of diagnosis, we conducted a review of these cases in the scientific literature. As mentioned before, primary bone lesions in RDD are rare, as confirmed by literature data. Most of the papers presented are very short series of case presentations.

Thus, we cite the study conducted by Andrew B. Ross [16] in which he describes two cases with primary, lytic lesions, located at the level of the distal radius epiphysis. The first case, a 20-year-old male, with proximal radius, the latter is the case of a 76-year-old woman, a rare case, because the disease is common in young men. The first case clinical symptoms were just elbow pain. The radiological aspect, expansile lithic lesion without sclerotic rim, with internal osseous septations, cortical thinning but no visible cortical breakthrough, leads to a differential diagnosis with metastasis, multiple myeloma, primary sarcoma of bone or atypical infectious process. CT and MRI exams were also conducted. Cortical thinning and small areas of cortical breakthrough not visible on the radiographs were apparent on the MRI. No associated soft tissue mass or perilesional edema was present. Histology examination demonstrated specific RDD features including emperipolesis (engulfment of intact lymphocytes contained with the cytoplasm of histiocyte cells) and positive S100 immunohistochemical staining. The second patient had a clinical history of intermittent left wrist pain. Noncontrast MRI showed multilocular, septated marrow replacing lesion in the distal radial metaphysis and epiphysis with heterogenous signal characteristics and without soft tissue mass or perilesional edema. As in the first case, the histological exam showed the same characteristic features of osseous RDD. Both cases were treated surgically, with curettage and bone graft, with favorable evolution.

These cases with features such as cortical thinning, focal extension into soft tissue, internal septation and, even periosteal reaction, were similar to other case reports, made by Arthur Vithran [20] or by Sundaram [21], features that resemble malignant neoplasia.

Another paper, by H J Grote [22], presents a case of a 33-year-old man with multiple primary bone involvement. Keith R. Bachmann [23] reports the case of a 71-year-old African American man, with a lytic lesion in the iliac wing, but with no clinical manifestations. In this case, conservative treatment was chosen. At a follow-up of 19 months the lesion was constant, without clinical manifestations. Jonathan C. Baker reports the case of a 19-year-old man with a primary lytic bone lesion, located in the distal femur. A CT-guided needle biopsy of the lesion was performed, with histological diagnosis showing chronic osteomyelitis. After a subsequent curettage and bone grafting, histological and immunohistochemical analysis leads to RDD diagnosis. The patient was subsequently evaluated by PET-CT examination for the evolution of the disease (RDD) [24]. Assessment and diagnosis by PET-CT examination of primary bone lesions in RDD were also reported by Jeffrey S. Tsang in a 41-year-old man [25].

Two larger series of primary bone lesions from the RDD have been reported in the literature. Elizabeth G Demicco [6] presents a series of 15 cases, eight women and five men aged between 3 and 56 years. Bone lesions, lytic with sclerotic margins have been found in multiple bone sites (tibia, femur, clavicle, skull, maxilla, calcaneus, phalanx, metacarpal and sacrum). The diagnosis was based on histopathological examination, classic features of RDD and consisted of a mixed inflammatory infiltrate with numerous large histiocytes with abundant eosinophilic cytoplasm which exhibited emperipolesis. Immunohistochemical stains showed that the large histiocytes were S-100 positive. The treatment was surgical, with curettage and bone grafting, with follow-up for 12 patients. Five of them developed over time, extraosseous manifestations, including testicular, lymph node and subcutaneous lesions. Another larger series of 14 cases, obtained in 10 years, was reported by R F Dong [26]. Eight men and six women between the ages of 2 and 64 were registered, showing lithic bone lesions with sclerotic margins in different bone sites. The diagnosis was based on histopathological examination, the lesions percolated through the medullary cavity in an infiltrative fashion and alternating hyper- and hypo-cellular regions of histiocytic clusters and large histiocytes also showed emperipolesis. Immunohistochemical staining showed that the large histiocytes were positive for S-100, CD68 and CD163 in all cases. The treatment was surgical, with curettage and bone grafting, the follow-up being performed for 12 cases. In total, 3 of the 12 patients experienced recurrences after the first surgery.

The data for this review are summarized in Table 2.

## 4. Discussion

Rosai–Dorfman’s disease is a rare condition characterized by reactive and polyclonal histiocytic proliferation, [16] due to its rarity having its own “R group” subtype in Non-Langerhans histiocytic disorders.

The exact etiology of RDD remains unknown at this time (idiopathic disease), infectious factors, herpes virus, parvovirus B19, Epstein Barr [27,28], or even bacterial infections [29] immunodeficiency or autoimmune diseases, being considered.

The typical clinical manifestations are represented by the involvement of the ganglion chains (painless swelling), especially the cervical ones, this being the classic “nodal” type. However, involvement of other tissues or organs, “extranodal involvement”, is not uncommon, with a frequency of 40%, often presenting as a subcutaneous mass [16]. Bone involvement in Rosai–Dorfman’s disease is rare, with a frequency of about 10%, usually occurring as a secondary process, in parallel with other organ involvement.

Radiological images, CT or MRI are not specific, and do not help in guiding the diagnosis, as occurred in our case. She presented with osteolytic lesions, at the level of the tibial shaft, with thinning and erosion of the bone cortex, in association with minimal periosteal reaction.

Histological examination is essential in guiding the diagnosis, emperipolesis being characteristic but variably present, and also immunohistochemical staining is very helpful, as the expression of S-100 is characteristic of RDD. However, this is often inconclusive, indicating non-specific inflammatory tissue as in the case of the first biopsy performed in our case [16].

These non-specific imaging features, the lack of disease characteristic symptomatology, together with inconclusive histopathological results, can represent diagnostic pitfalls, leading to confusion.

The case reported is interesting for atypical clinical evolution, not specific to a chronic inflammatory disease, with the extraosseous development of the diaphyseal tumor. This manifestation accentuates the diagnostic confusion, being necessary to be careful for differential diagnosis with more aggressive bone diseases, such as adamantinoma, xanthoma, osteolytic osteosarcoma, or osteolytic bone metastases, as we presented in the case report. At the same time, the clinical aspects, laboratory markers, CT and MRI and radiological imaging, must be carefully evaluated in these cases to avoid confusion with the mentioned diseases.

Repeating the biopsy in these difficult to diagnose cases, is mandatory. As we have noted, and as mentioned in the literature [16], percutaneous biopsies with biopsy needle can induce inconclusive histopathological results (non-specific inflammatory tissue) while excisional biopsies with multiple tissue samples from different areas of the tumor can help to obtain a correct histopathological result. Histopathological examination was essential for diagnosis, requiring increased attention to observe the phenomenon of emperipolesis, while immunohistochemical staining has identified expression of S-100 intensely positive in macrophage/xanthoma type cells, which is characteristic of RDD.

Due to the rarity of this disease, before initiating surgical treatment, it is recommended to correlate the data obtained (clinical, laboratory, imaging, histological) with data from the scientific literature, for more reliable certification of the obtained diagnosis. The choice of surgical treatment, block resection and reconstruction or curettage and grafting of the remaining bone defect depends on the local evolution and the degree of destruction of the bone tissue. Curettage and grafting are the most used, as we have performed. In addition, we recommend safety osteosynthesis, with plate and screws as an additional measure, to avoid a fracture by eroding the bone cortex.

RDD primary bone involvement is in fact an unusual manifestation of a rare disease. The fact that they are very rare in clinical practice, associated with nonspecific imaging and a potential inconclusive histopathological result, especially in the case of needle biopsies, leads to frequent diagnostic errors.

This case was brought to attention especially for the atypical evolution, presumably as a chronic inflammatory lesion, especially to point out the importance of the correct diagnosis that, in this particular situation, could have had a totally unsatisfactory outcome.

## Figures and Tables

**Figure 1 diagnostics-12-00783-f001:**
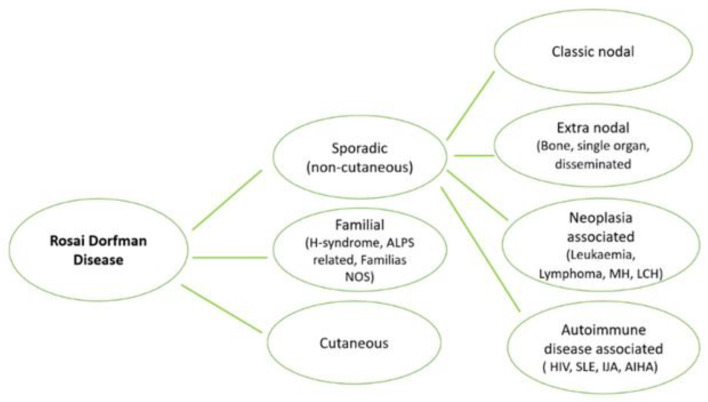
Addapted from Histiocyte Society, Rosai–Dorfman classification [3]. AIHA, autoimmune hemolytic anemia; ALPS, autoimmune lymphoproliferative syndrome; ECD, Erdheim–Chester disease; IJA, idiopathic juvenile arthritis; LCH, Langerhans cell histiocytosis; MH, malignant histiocytoses; NOS, not otherwise specified; SLE, systemic lupus erythematosus.

**Figure 2 diagnostics-12-00783-f002:**
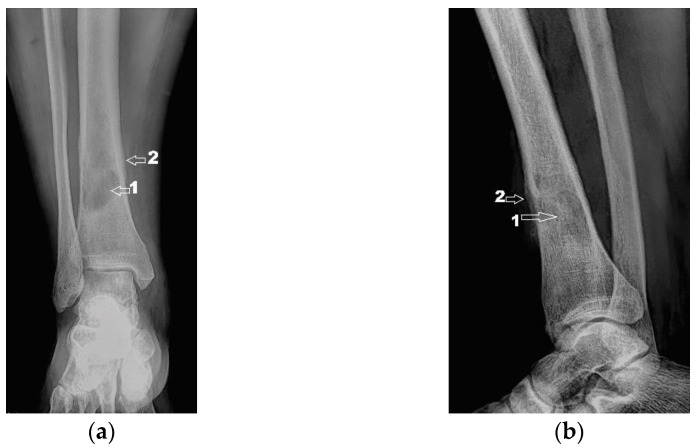
(**a**) Radiological image (coronal view); (**b**) Radiological image (lateral view). 1—osteolytic lesion; 2—periosteal reaction.

**Figure 3 diagnostics-12-00783-f003:**
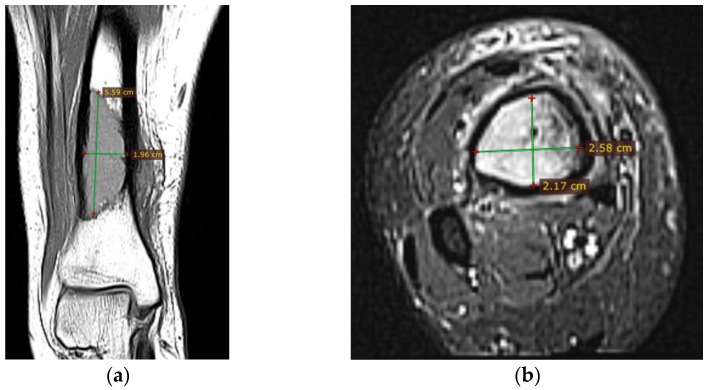
(**a**) MRI T1 coronal section, hyposignal lesion; (**b**) MRI T1 fat sat, axial section.

**Figure 4 diagnostics-12-00783-f004:**
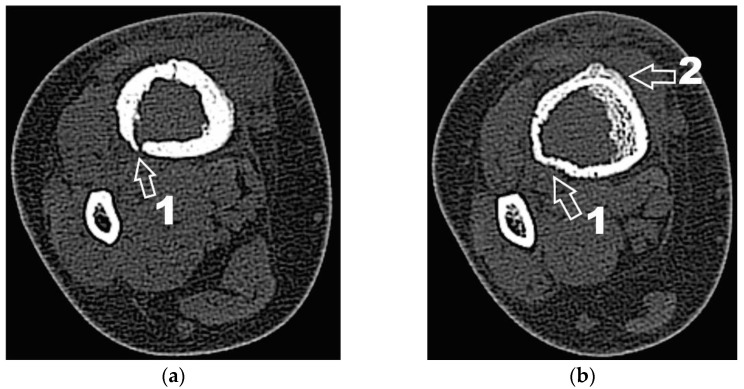
(**a**) CT axial section. 1—disruption of the bone cortex; (**b**) CT axial section. 1—thinning of bone cortex, 2—periosteal reaction.

**Figure 5 diagnostics-12-00783-f005:**
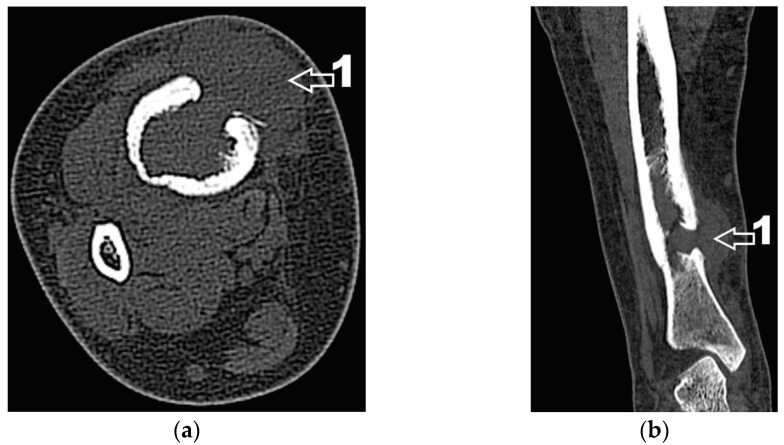
(**a**) CT axial section. 1—extraosseous expansion of the tumor formation; (**b**) CT coronal section. 1—extraosseous expansion of the tumor formation.

**Figure 6 diagnostics-12-00783-f006:**
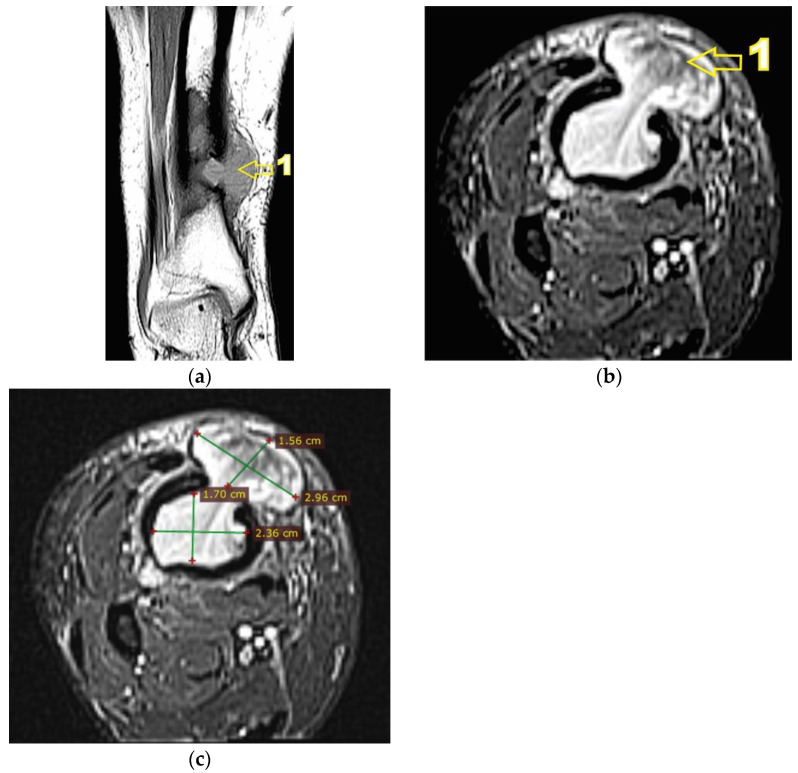
(**a**) MRI T1 hyposignal coronal section. 1—extraosseous expansion of the tumor formation; (**b**) MRI T2 Fat Sat (STIR) axial section. 1—extraosseous expansion of the tumor formation; (**c**) MRI T2 Fat Sat (STIR) axial section, tumor dimensions.

**Figure 7 diagnostics-12-00783-f007:**
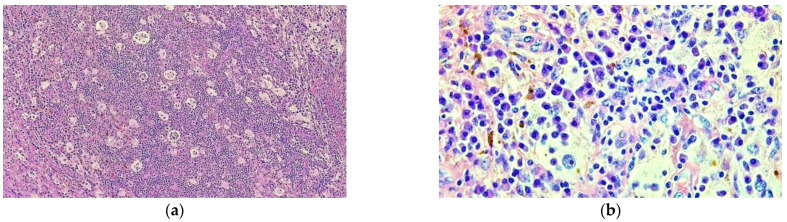
Prominent population of histiocytes characterized by large, round nuclei and voluminous cytoplasm showing emperipolesis with engulfment of lymphocytes (**a**) HE staining; 40×; (**b**) HE staining 400×.

**Figure 8 diagnostics-12-00783-f008:**
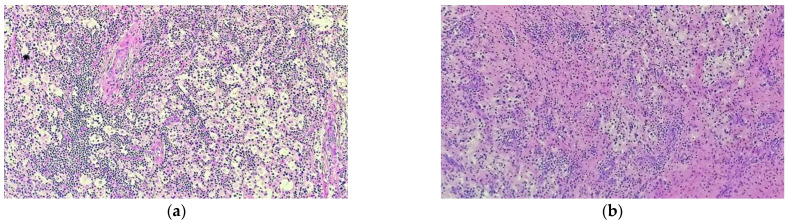
Rosai–Dorfman disease showing (**a**) a prominent histiocytic infiltrate in a background of inflammatory cells, predominantly comprised of lymphocytes and plasma cells (HE staining 40×); (**b**) lesional histiocytes associated with fibrosis and prominent inflammatory infiltrate extended into adjacent bone and soft tissue (HE staining, 40×).

**Figure 9 diagnostics-12-00783-f009:**
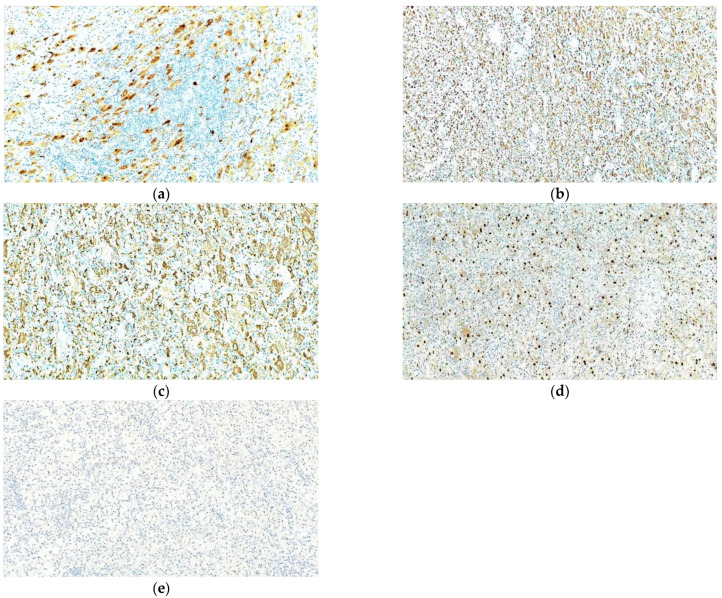
Rosai–Dorfman disease of the bone. Immunohistochemical positive reactions for (**a**) S100; (**b**) CD68; (**c**) CD163; (**d**) Cyclin D1; (**e**) negative for CD1a; in the lesional histiocytes. IHC staining with DAB chromogen, 100×.

**Figure 10 diagnostics-12-00783-f010:**
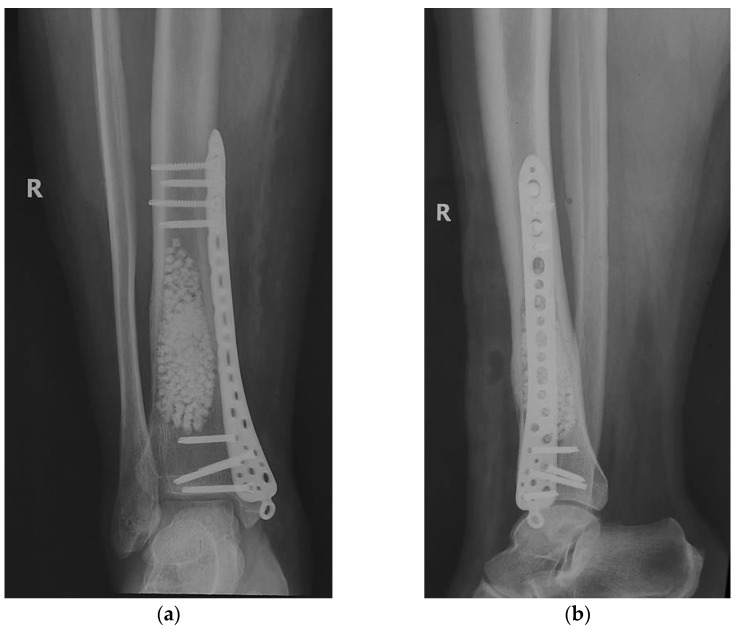
Curettage, bone substitute grafting and plate osteosynthesis. After surgery radiographs (**a**) Radiological, coronal view; (**b**) Radiological, lateral view.

**Figure 11 diagnostics-12-00783-f011:**
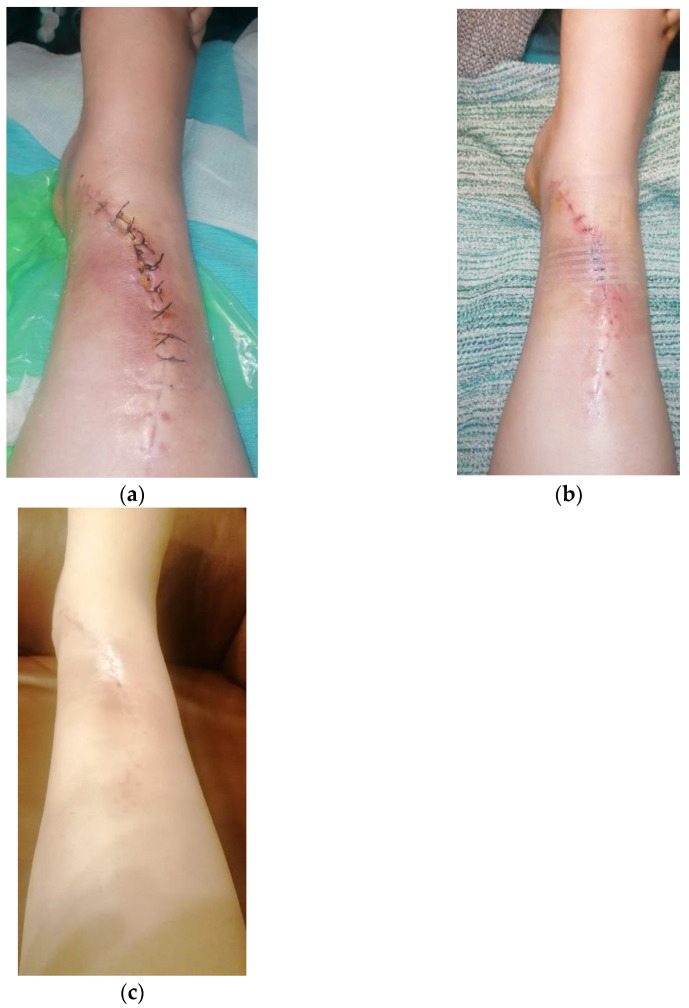
Skin dehiscence evolution: (**a**) Primary dehiscence; (**b**) Per secundam suture; (**c**) Healed wound.

**Figure 12 diagnostics-12-00783-f012:**
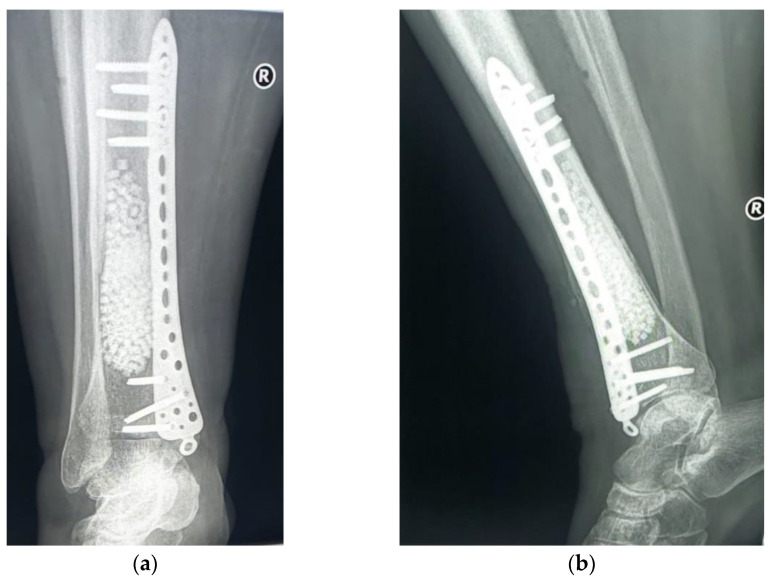
Radiological 6-month follow-up: (**a**) coronal view; (**b**) lateral view.

**Table 1 diagnostics-12-00783-t001:** Differential diagnosis features.

Disease	Radiological and Imagistic (CT, MRI) Features	Histopathology	Alkaline Phosphatase	Inflammatory Markers
Adamantinoma	Osteolytic lesions with sclerotic rim; thinning of the bone cortex, no periosteal reaction; rare extraosseous involvement (recurrent lesions)	Epithelial and osteofibrous components; uniformly positive immunohistochemy for keratins 14 and 19 (CKs 5, 14 and 19)	Within normal limits	Within normal limits
Xanthoma	Pseudotumor osteolytic lesion; erosion of the bone cortex; no periosteal reaction	Xanthomatous cells identified in a conjunctive tissue	Within normal limits	Within normal limits
Osteolytic osteosarcoma	Aggressive osteolytic lesion; thinning and disruption of the bone cortex; extraosseous development; periosteal reaction; not well defined extraosseous development and without peripheric capsule.	Specific cells	Increased levels	Inconstant increased levels
Osteolytic bone metastases	Osteolytic lesions; thinning and disruption of the bone cortex; no periosteal reaction; extraosseous extension is not well defined and does not have a capsule at the periphery	Specific cells	Increased levels	Within normal limits
Extranodal Rosai–Dorfman Disease	Lytic lesions with well-defined sclerotic margins; pure sclerotic lesions are rare; cortical thinning and focal breakthrough are common findings	Emperipolesis (engulfment of intact lymphocytes contained with the cytoplasm of histiocyte cells); positive S100 immunohistochemical staining.	Within normal limits	Increased levels (not mandatory)
	Periosteal reaction is rare; sometimes internal osseous septations can be identified; soft tissue mass in continuity with the bone lesion is a rare finding			

**Table 2 diagnostics-12-00783-t002:** Literature review, number of cases, bone site, treatment and type of disease.

Authors/Ref	No. of Cases	Bone Involvement	Treatment	Classification
Andrew B. Ross [16]	Two	Proximal radius Distal radius	Curettage and bone grafting	Extranodal
Arthur Vithran [20]	One	Tibia	Curettage and bone grafting	Extranodal
Sundaram [21]	One	Distal femur Fibula	Curettage and bone grafting	Extranodal
H J Grote [22]	One	Radius Tibia	Curettage and bone grafting	Extranodal
Keith R. Bachmann [23]	One	Iliac wing	Conservative treatment	Extranodal
Jonathan C. Baker [24]	One	Distal femur	Curettage and bone grafting	Extranodal
Jeffrey S. Tsang [25]	One	Tibia	Curettage and bone grafting	Extranodal
Elizabeth G Demicco [6]	15	Tibia, femur, clavicle, skull, maxilla, calcaneus, phalanx, metacarpal, sacrum	Curettage and bone grafting	Extranodal Nodal
R F Dong [26]	14	Tibia, fibula, femur, skull, radius	Curettage and bone grafting	Extranodal Nodal

## Data Availability

Not applicable.

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
