# Peer review of "Primary Bone Lesions in Rosai–Dorfman Disease, a Rare Case and Diagnostic Challenge—Case Report and Literature Review"

_diagnostics, 2022, doi:10.3390/diagnostics12040783_

Round 1

Reviewer 1 Report

The authors report a rare case of extra-nodal RDD arising in the bone. The patient was managed well. The case report is well written, and the literature review is adequate. However, it is advisable that the authors provide more details of the previously published cases/ series, and a table in this regard will be useful. The paper also needs editing for grammar and syntax. 

Author Response

Hello sir

Thanks for your review. Following your comments, we have modified the case presentation as follows:
- I added additional data regarding the cases from the scientific literature and I made a table in which I highlighted the number of cases, bone site, treatment, and type of disease
- the paper was revised by a native English speaker, in terms of grammar and syntax

Sincerely yours

Razvan Adam

Reviewer 2 Report

The case presented is of interest.

The article needs english revision by a native speaker.

Specific suggestion:

  • Please avoid the use the first person (e.g. "I did not agree" used in the abstract must be changed); this doesn't sound scientific.
  • In MRI figure legend, please add the sequences shown
  • Replace Front with Coronal view (you used both in the paper)
  • I suggest to start with the case report, afterwards perform the literature reviews. Please, follow the sequence you used in the abstract to draw the paper.
  • Please, underline better which are the clinical and radiological features that may lead to Rosai-Dorfman suspicion.
  • Adding some tables could be useful (e.g. differential radiological diagnosis, different disease characteristics...)

Author Response

Hello sir

Thanks for your review. Following your comments, we have modified the case presentation as follows:

  • The use the first person was eliminated from text
  • The MRI sequences used were added in the legend of the figures 
  • Front view was replaced with coronal view in radiology figure legend
  • Literature review chapter was moved after the case presentation, as you suggested 
  • We have added additional data on imaging radiological and histology features that led to the diagnosis of RDD
  • A table with differential diagnosis features, such as radiological and imagistic (CT, MRI) features, histology, lab markers, was added
  • The paper was revised by a native English speaker, in terms of grammar and syntax
    Sincerely yours
    Razvan Adam

Round 2

Reviewer 1 Report

The paper may be accepted 

Reviewer 2 Report

I appreciate the revisions performed,

Thank you for the work done!